# Conformational Analysis of 1,5-Diaryl-3-Oxo-1,4-Pentadiene Derivatives: A Nuclear Overhauser Effect Spectroscopy Investigation

**DOI:** 10.3390/ijms242316707

**Published:** 2023-11-24

**Authors:** Konstantin Belov, Valery Brel, Valentina Sobornova, Irina Fedorova, Ilya Khodov

**Affiliations:** 1G.A. Krestov Institute of Solution Chemistry, Russian Academy of Sciences, 153045 Ivanovo, Russia; kvb@isc-ras.ru (K.B.); vvs@isc-ras.ru (V.S.); fiv@isc-ras.ru (I.F.); 2A.N. Nesmeyanov Institute of Organoelement Compounds, Russian Academy of Sciences, 119334 Moscow, Russia; v_brel@mail.ru

**Keywords:** NMR, structure, NOESY, conformation

## Abstract

1,5-Diaryl-3-Oxo-1,4-Pentadiene derivatives are intriguing organic compounds with a unique structure featuring a pentadiene core, aryl groups, and a ketone group. This study investigates the influence of fluorine atoms on the conformational features of these derivatives in deuterated chloroform (CDCl_3_) solution. Through nuclear magnetic resonance (NMR) spectroscopy and quantum chemical calculations, we discerned variations in interatomic distances and established predominant conformer proportions. The findings suggest that the non-fluorinated entity exhibits a uniform distribution across various conformer groups. The introduction of a fluorine atom induces substantial alterations, resulting in the predominance of a specific conformer group. This structural insight may hold the key to their diverse anticancer activities, previously reported in the literature.

## 1. Introduction

A significant issue associated with using anticancer pharmaceutical ingredients for treating cancer of different origins is the broad array of adverse effects, including hematological toxicities, neurotoxicity, and cardiotoxicity [1,2,3,4,5]. Chemotherapeutic drug molecules impact various body parts, including the skin, hair, bone marrow, blood, lungs, gastrointestinal tract, and kidneys [6,7,8,9]. A practical approach to address such issues is the utilization of naturally derived chemicals. Restrepo and co-workers note that the natural world provides a limitless reservoir of various chemical compounds with diverse structures [10].

Consequently, multinational initiatives like the National Cooperative Drug Discovery Groups (NCDDGs) have been established to identify anticancer drugs derived from nature [11]. Biological substances, such as those derived from plants, microorganisms, and animals that live in the sea or on land, have demonstrated promise in creating medications that can combat cancer [11]. In addition, as Sofi and co-workers note in their work [12], using semisynthetic methods to modify natural products has resulted in the development of novel anticancer medicines that exhibit enhanced therapeutic effectiveness and low adverse reactions.

Curcumin, a naturally occurring molecule derived from the root of the perennial herb turmeric, is considered a potential compound for preventing and treating cancer [13]. Curcumin hinders cellular growth, restrains invasion and movement, stimulates programmed cell death, triggers the self-degradation of cells, diminishes inflammation, and governs diverse cellular mechanisms in the advancement of cancer [14]. Curcumin is considered safe and well tolerated, even when administered at high dosages [15]. However, the effectiveness of curcumin in treating cancer is limited due to its restricted absorption and utilization by the body [16]. Nevertheless, to achieve the best clinical effectiveness, it is crucial to tackle the issues related to its bioavailability and formulation [14]. The study conducted by Plummer and co-workers [17] has demonstrated the alteration of curcumin to maintain its efficacy and improve its ability to be absorbed by the body [18].

1,5-diaryl-3-oxo-1,4-pentadienes, which are variations of curcumin, show potential for being modified and used to develop new cytostatic chemicals [18]. The study by Shibata H. et al. [18] focuses on determining the half maximum inhibitory concentration (IC_50_) for 86 variations of 1,5-diaryl-3-oxo-1,4-pentadienes, including both linear and cyclic structures. The authors demonstrated the superior efficacy of certain 1,5-diaryl-3-oxo-1,4-pentadiene derivatives compared to curcumin. The modification of compounds primarily entailed the incorporation of different substituents into the para locations of the aryl fragments of the molecules. The study conducted by Brel et al. [19] presents an alternative method for altering the composition of these molecules. The cornerstone of this study involves the cyclic alteration of 1,5-diaryl-3-oxo-1,4-pentadiene, where functional groups are substituted by adding piperidone through the nitrogen atom. The literature presents fragments of N-aryl fumaraminic acids [20], amino acids [21], and nitroxides [22] as substituents. Brel et al. [19] employ an organophosphorus moiety, specifically diethyl(phenyl)methylphosphonate, as a substituent. The researchers discovered that the phosphonate derivatives of the 3,5-bis(arylidene)-4-piperidone series showed inhibitory effects on specific human cancer cells (RD, PC3, HCT116, and MCF7). The IC_50_ values for these derivatives ranged from 2.5 to 8.5 µM [19].

Nevertheless, the authors acknowledge that additional progress in this field necessitates investigations into mechanistic elements, such as spatial organization and intermolecular interactions. We were motivated by this comment to conduct initial investigations into the spatial arrangement and flexibility of two 1,5-diaryl-3-oxo-1,4-pentadiene compounds. The findings from these experiments are detailed in this publication.

Nuclear Overhauser effect spectroscopy (NOESY) is a powerful technique to ascertain the spatial structural characteristics of active pharmaceutical ingredients (APIs) in small molecules [23,24,25,26]. The utilization of NOESY in the study conducted by Khodov and co-workers [27] has enabled the acquisition of precise quantitative data regarding the prevailing molecular conformation in a solution. This knowledge is essential for the strategic development and execution of synthesizing novel chemicals and altering pre-existing ones. Furthermore, as elucidated in the article by Scheidt et al. [28], the configuration of a molecule can have a crucial impact on its ability to attach to the target molecule, thereby influencing its biological functionality. Nevertheless, investigating 1,5-diaryl-3-oxo-1,4-pentadiene derivatives necessitates a unique technique due to the molecule’s inherent conformational instability, leading to numerous conformers.

This study conducted a comprehensive analysis of two derivatives of 1,5-diaryl-3-oxo-1,4-pentadiene. One derivative contains a fluorine atom in the para position of the cyclic fragment of diethyl(phenyl)methylphosphonate (compound **2**). In contrast, the other derivative does not contain a fluorine atom (compound **1**). We will present various methodologies to analyze these compounds to establish the prevailing conformations of the studied objects. The acquired knowledge will be valuable for advancing the development of the field related to synthesizing novel therapeutic compounds using 1,5-diaryl-3-oxo-1,4-pentadiene as a basis. Additionally, it will facilitate advancements in comprehending the mechanisms by which these compounds interact with target molecules.

## 2. Results and Discussion

1,5-Diaryl-3-Oxo-1,4-Pentadiene derivatives are chemical compounds that have a unique structure. They consist of a pentadiene core with two conjugated double bonds (diene) at the first and fifth positions. The framework has two aryl groups, augmenting stability and reactivity. A ketone group (C=O) at the third carbon atom adds an oxygen atom and influences the chemical characteristics. The features of these derivatives may vary based on the substituents present in the aryl groups, leading to a diverse variety of properties. The inquiry examines the impact of fluorine atoms on the conformational features of compound **2**, thereby affecting their qualities and potential for therapeutic uses. This work aimed to determine the proportions of various conformer groups in 1,5-diaryl-3-oxo-1,4-pentadiene derivatives, specifically compound **1** and compound **2**, when dissolved in the organic solvent CDCl_3_. The aim was to examine the influence of fluorine on the existing molecular conformation. Compound **1** consists of a 3,5-dibenzylidene-4-oxopiperidine structure (red region) connected to diethyl(phenyl)methylphosphonate structure (green region) through the piperidone nitrogen atom (refer to Figure 1a). Compound **2** exhibits a distinct structural variation from compound **1** due to the inclusion of fluorine in the phenyl radical (highlighted in blue) (refer to Figure 1b). The alterations in the arrangement of the molecules in the objects being examined can be described by the variations in the dihedral angles τ_1_ (C_6_-N_1_-C_7_-C_8_) and τ_2_ (C_2_-N_1_-C_7_-P_1_), which are a result of the rotation of the diethyl(phenyl)methylphosphonate. It should be observed that structural flexibility comes from the ability to change dihedral angles of the diethyl(phenyl)methylphosphonate moiety within the molecule fragment. 

Nonetheless, alterations in these angles are constrained to such an extent that they fall below the detectable threshold attainable through the employed NOESY techniques. The angle τ_1_ (C_6_-N_1_-C_7_-C_8_) values for compound **1** and compound **2** are highly similar. In Figure 2a, the black dots correspond to the conformers of compound **1**, while the red dots represent the conformers of compound **2**. It is observed that, in the majority of situations, both the energy and angle values are identical. The primary deviations from this pattern can be attributed to the limited number of computed conformers with energies below 36.35 kJ/mol for compound **2** (only 39 structures), compared to the more extensive set of 62 structures obtained for compound **1**. Figure 2a displays graphs that reveal the presence of two distinct groups of conformers. These groups may be identified based on their angle values, which range from −55° to −150° and from 60° to 150°. Figure 2b demonstrates a comparable pattern at the angle τ_2_ (C_2_-N_1_-C_7_-P_1_).

To determine the probable conformation of the molecules of the compounds under study, ^1^H spectra were analyzed using the NMR method. The assignment of resonance signals in the obtained spectra corresponding to hydrogen atoms in the structure of the molecules (see Figure 3) was performed using techniques such as ^13^C (Appendix A) ^1^H–^13^C HSQC (Appendix A), ^1^H–^13^C HMBC (Appendix A), and ^1^H–^1^H TOCSY (Appendix A). The analysis made it possible to establish correlations between hydrogen and carbon atoms connected by one (^1^H–^13^C HSQC) [29] or more (^1^H–^13^C HMBC) chemical bonds [30,31], as well as interactions between hydrogens in a continuous chain of chemical bonds (^1^H–^1^H TOCSY) [32,33,34].

Upon analyzing the chemical shifts of NMR signals in the ^1^H spectra, it was seen that there are two signals in the high field region corresponding to the methyl groups of diethyl(phenyl)methylphosphonate (H23 and H24) [35]. Within the frequency range of 3 to 5 ppm, there are a total of seven NMR signals associated with the CH_2_ groups of the diethyl(phenyl)methylphosphonate (H21a, H21b, H22a, and H22b) and oxypiperidine (H2 and H6) fragments [36,37]. Additionally, there is a signal for the CH group (H7) bonded to the nitrogen atom of piperidine. The region between 6 and 8 ppm exhibits NMR signals corresponding to the hydrogen atoms of the phenyl (H10/12, H9/13; H11—for compound **1**) and dibenzalidene (H14, H16/20, H17/19, and H18) fragments of the molecules [38].

When comparing the ^1^H NMR spectra of diethyl(phenyl)methylphosphonate compounds, it is essential to observe that the signals of the methyl groups are closer together in compound **2**. In a precise manner, the H24 signal experiences a 0.02 ppm shift towards the low magnetic field region. In contrast, the H23 signal undergoes a 0.01 ppm shift towards the high magnetic field region. The hydrogen atoms, situated within the range of 3 to 5 ppm, exhibit a minor shift in their location when transitioning from compound **1** to compound **2**. Therefore, in comparison to compound **1**, compound **2** exhibits a shift of the H22b proton signal by 0.01 ppm towards the weak field region. The chemical shifts of signals H22a and H21b remain unchanged, while signals H6 and H21a experience a shift of 0.02 ppm towards the weak field region.

Conversely, signals H2 and H7 shift to 0.02 ppm towards the high field region. The chemical shifts of the signals associated with the phenyl and dibenzylidene segments see the most significant alterations. Compound **2** has a downfield shift of 0.33 ppm for the H10/12 signal, corresponding to hydrogen atoms in the meta position of the phenyl fragment. Conversely, the H9/13 signal, representing protons in the ortho position, experiences an upfield shift of 0.05 ppm. Regarding the signals of the dibenzalidene fragment, the H14 and H18 signals are displaced towards the higher chemical shift region by 0.01 ppm and 0.08 ppm, respectively.

On the other hand, the H16/20 and H17/19 signals are shifted towards the lower chemical shift region by 0.01 ppm and 0.33 ppm, respectively. The H11 signal is exclusively detected in the ^1^H NMR spectrum of compound **1**. This is because the hydrogen atom at the para position of the phenyl fragment of compound **2** is substituted with a fluorine atom. The primary factor influencing the changes in the chemical shift of signals in the ^1^H NMR spectra of two compounds, which only differ in the substituent located in the para position of the phenyl fragment of the molecule, and are measured in identical solvents, is mainly attributed to the screening effect [39]. The alteration in shielding is linked to three primary factors: the incorporation of a fluorine atom as a substituent, a modification in the likely conformation of the molecule, and the impact of ring currents originating from the phenyl fragment [40,41,42,43], which is also a result of the conformational change. The observed NMR signals do not exhibit a distinct inclination to shift towards a strong or weak magnetic field. The alteration in their chemical shifts results from the simultaneous influence of two competing processes: the insertion of a substituent and a modification in the molecular conformation [44]. The most significant disparity in chemical shift values is detected for signals in the low magnetic field region (phenyl and dibenzylidene fragments). This discrepancy arises due to substituting a fluorine atom for a proton in the para position of the phenyl fragment in compound **2**. The rotation of the diethyl fragment in the molecule causes a change in its conformation, resulting in a shift in the position of the phenyl ring relative to the dibenzylidene fragment of the molecules being studied [45]. This shift is responsible for the impact of ring currents [40,46].

Examining chemical shifts of signals in ^1^H NMR spectra enabled us to identify cross-peaks in NOESY spectra (refer to Figure 4) at various mixing periods (τ_m_), ranging from 0.15 s to 0.75 s with an increment of 0.05 s. The NOESY spectra consist of a two-dimensional map where the projections of ^1^H NMR spectra are positioned on both axes. The NOESY spectra display diagonal signals at coordinates ii, jj, etc., which correlate to the signals observed in the ^1^H NMR spectra. Additionally, cross-peaks at coordinates ij, ji, etc., indicate the interaction between hydrogen atoms in the molecular structure of the studied substances. Cross-peak NOESY indicates the presence of hydrogen atom interactions within a range of around 5–6 Å [47].

The analysis of the NOESY spectra revealed the presence of 18 cross-peaks in both compound **1** and compound **2.** Some of these cross-peaks are shared between the two compounds, while others are unique to each compound. It is essential to mention that the NOESY spectrum of compound **2** (Figure 4b) does not show any cross-peaks indicating interactions with the proton (H11) of the phenyl ring of diethyl(phenyl)methylphosphonate in the para position. For instance, there is no cross-peak between H7 and H11. The fluorine (F) atom is situated at the specified position, and its signals are not present in the proton (^1^H) spectrum of compound **2**.

Furthermore, the NOESY spectrum for compound **1** (Figure 4a) exhibits the H9/13-H23 cross-peak, which indicates the distance between the protons in the ortho position of the phenyl ring and the CH_3_ group of diethyl(phenyl)methylphosphonate. However, this cross-peak is not found in the NOESY spectrum for compound **2**. The probable cause of this condition is the overlapping of signals in the low-intensity area for compound **2**, as previously mentioned. The H18-H23 cross-peak is exclusively detected in the NOESY spectra of compound **2**. It represents the distance between the proton of the phenyl ring in the dibenzylidene fragment and the CH_3_ group in diethyl(phenyl)methylphosphonate. A conformational change in the molecule most likely causes the disappearance of the cross-peak in the NOESY spectrum of compound **1**. When diethyl(phenyl)methylphosphonate is rotated, the distance associated with the discussed cross-peak rises, preventing its detection. The final discrepancy in the observed cross-peaks in the NOESY spectra (Figure 4) can be attributed to the absence of the H2–H7 cross-peak in the NOESY spectrum of compound **1** (Figure 4a). Conversely, this cross-peak is observed in compound **2**, most likely due to the overlapping cross-peaks of H2-H7 and H2-H6 in the NOESY spectrum of compound **1** (Figure 4a).

In the study conducted by the authors of [48], they introduced a reference-free nuclear Overhauser effect (NOE) methodology for determining the predominant conformation of small molecules utilizing NOE spectroscopy data. Specifically, the authors computed normalized integrated intensity values of cross-peaks observed in NOESY spectra and subsequently derived the corresponding internuclear distances based on molecular mechanics data for individual conformers. By analyzing graphs depicting the relationship between normalized integral intensity and internuclear distance, coefficients of determination (R^2^) were calculated for each studied object’s structure. These values provided valuable insights into the likely conformations of the subject compounds.

This approach demonstrated efficacy in cases of rigid molecular structures. However, it exhibited limitations when applied to conformationally labile structures. Nevertheless, this method remained valuable for selecting suitable reference and conformation-dependent distances.

In our current investigation concerning compound **1** and compound **2**, we adopted a similar approach, as proposed by the authors of [48], but with a modification. Instead of relying on normalized integral intensities, we employed cross-relaxation rates. This adaptation is deemed more appropriate, as cross-relaxation rates offer a direct correlation with both internuclear distances (as described in Equation (1)) and the conformation of molecules in solution (as illustrated in Equation (2)), as evidenced in the work of Lee and Krishna [49].
(1)$$\sigma_{ij}=\frac{1}{r_{ij}^{6}}$$
(2)$$\sigma_{ij}=\sum_{i}\sigma_{i}P_{i}$$
where *σ_ij_* is the cross-relaxation rate between nuclei *i* and *j*, and *r_ij_* is the distance between them. *σ_i_* is the cross-relaxation rate in the conformer *i*, and *P_i_* is the relative fraction of this conformer.

All detected cross-peaks and diagonal signals present in the NOESY spectrum underwent integration to ascertain the normalized values of integrated cross-peak intensities, employing the Peak Amplitude Normalization for an Improved Cross-relaxation (PANIC) model [50,51,52], as depicted in Equation (3).
(3)$$I_{ij}^{\exp}\left(\tau_{m}\right)=\frac{1}{2}\left(\frac{1}{n_{j}}\left|\frac{a_{ij}\left(\tau_{m}\right)}{a_{ii}\left(\tau_{m}\right)}\right|+\frac{1}{n_{i}}\left|\frac{a_{ji}\left(\tau_{m}\right)}{a_{jj}\left(\tau_{m}\right)}\right|\right)$$
where *n_j_* and *n_i_* are the parameters indicating the number of protons that determine the contribution to the integrated intensity of the cross-peak, *a_ij_* and *a_ji_* are the parameters that determine the intensity of cross-peaks in the 2D NOESY spectra, and *a_ii_* and *a_jj_* are the parameters that determine the intensity of the diagonal signals in the two-dimensional spectra.

The following works [53,54,55,56] have demonstrated that the use of the PANIC approach facilitates linearizing the relationship between integrated cross-peak intensity and mixing time, as illustrated in Appendix A. This methodology proves instrumental in expanding the dataset by employing more extended mixing times, thereby enhancing the precision of cross-relaxation rate determination within the initial rate approximation (IRA) method [57,58,59,60].

The concurrent utilization of the PANIC method and the IRA method has enabled the precise determination of cross-relaxation rates corresponding to distinct internuclear distances employing NOESY spectral data. Subsequently, the internuclear distances associated with cross-peaks in the NOESY spectrum were computed utilizing conformer structures derived from quantum chemical calculations.

Specialized averaging models were employed since many of these calculated distances involve interactions with CH_2_ and CH_3_ groups. These models account for intramolecular lability, characterized by the rotational correlation time and the rate of molecular group motion, categorized as slow (>100 ps), medium-speed (50–100 ps), and fast (<50 ps) [61].

In these calculations, the DIST_ACESS program (RU 2020618574), a component of the Unified Register of Russian Programs for Electronic Computers and Databases, was utilized.

In order to calculate the distances involving molecular fragments with slow types of motion, such as the flipping of proton atoms of the CH groups in the benzene ring (H9/13-H16/20) [62], the averaging model can be applied using Equation (4):(4)$$r^{eff}={\left[\frac{1}{n_{I}n_{S}}\sum_{i}\frac{1}{r_{i}^{6}}\right]}^{-1/6}$$

Equation (5) determines the distance values for chemical groups, such as CH_2_ groups (H6–H9/13; H14–H16/20; H2–H16/20), that have an average speed of motion:(5)$$r^{eff}={\left[{\left(\frac{1}{n_{I}n_{S}}\sum_{i}\frac{1}{r_{i}^{3}}\right)}^{2}\right]}^{-1/6}$$
where *r^eff^* is the averaged internuclear distance taken from the conformer structures obtained in quantum-chemical calculations, *n_I_* and *n_S_* are the numbers of equivalent spins in the atom groups *I* and *S*, and *r_i_* is the distance between the spins in the considered groups.

This model is suitable for ascertaining distances within the CH–CH_2_, C_6_H_5_–CH_2_, and CH–CH_3_ groups. Nevertheless, in the case of methyl groups characterized by rapid motion, as previously indicated in [63], Equation (5) offers only an approximate solution. For the calculation of distances involving methyl groups, it is advisable to utilize Equation (6):(6)$$r^{eff}={\left[\frac{1}{5}\sum_{k=-2}^{2}\left|\frac{1}{3}\sum_{i=1}^{3}\frac{Y_{2k}\left(\theta_{mol}^{i},\varphi_{mol}^{i}\right)}{r_{i}^{3}}\right|\right]}^{-1/6}$$
where *θ_mol_* and *φ_mol_* are the polar angles of the internuclear vector in the molecular reference frame, and *Y_2k_* are the second-order spherical harmonics.

The data on internuclear distances and cross-relaxation rates were used to establish the relationship shown in Figure 5. Equivalent graphs were obtained for every conformer of compound **1** (refer to Appendix A) and compound **2** (refer to Appendix A). The data acquired from the graph were approximated using the reference-free model described [48].

The determination coefficients resulting from the data analysis of individual conformers of the studied objects are graphically presented in Figure 6. Figure 6 illustrates that discerning a predominant conformation is challenging due to the closely comparable determination coefficients among multiple conformers. This proximity in coefficient values can be attributed to the substantial conformational flexibility inherent to the molecules under investigation.

Figure 7 illustrates a comprehensive plot encompassing all conformers, revealing notable disparities in the magnitudes of specific interatomic distances. This variation subsequently results in analogous R^2^ values across distinct conformer structures.

In Figure 7, it is evident that specific distance values remain unvaried during conformational changes, holding a consistent position along the abscissa axis (referred to as reference distances—H22a–H22b, H9/13–H10/12, H23–H21a, H22a–H24, H22b–H24, H14–H2). Conversely, other distances exhibit substantial variation as conformational changes occur, resulting in a wide range of values (termed experimentally determined distances—H7–H9/13, H6–H7, H2–H16/20, H14–H16/20, H2–H9/13, H6–H9/13).

It is worth noting that a third category of distances, which surpasses 5 Å, is depicted in gray and has been deemed unsuitable for determining the proportions of conformer groups for the subjects under investigation in this study. Furthermore, it is essential to highlight that, for clarity, Figure 7 does not encompass all computed distances but only presents a representative selection. Consequently, distances such as H9/13–H24, H9/13–H23, and H16/20–H23 are omitted from Figure 7. This omission is justified as the positions of these distances substantially overlap with most data points corresponding to distances characterized by low cross-relaxation rates, which would impede the visualization and interpretation of results.

In order to ascertain the intermolecular distances of the investigated molecules, the isolated spin pair approximation model (ISPA) method was employed, as outlined in Equation (7) (Appendix A):(7)$$r_{\exp}=r_{0}{\left(\frac{\sigma_{0}}{\sigma_{\exp}}\right)}^{1/6}$$

Hence, provided that all experimental parameters about the cross-relaxation rate (σ_0_ and *σ*_exp_) have been accurately ascertained, interproton distances can be determined with notable efficiency. It becomes feasible to compute the experimental distance, denoted as *r*_exp_, by merely knowing a single distance, typically represented as the reference distance *r*_ref_. A comprehensive elaboration of this approach can be found in previous publications [57,64,65]. The figures presented in Figure 7 facilitated the selection of appropriate experimentally determined (H6–H9/13) and reference (H14–H2) distances. The H6–H9/13 distance exhibited considerable variability, reflecting distinct molecular conformations, further classified into two groups, denoted as A and B (see Figure 8 and Appendix A).

For compound **1**, these values were observed as 3.07 ± 0.30 Å (*r_A_*) and 4.48 ± 0.32 Å (*r_B_*). Compound **2** displayed analogous variations, with H6–H9/13 distances of 3.40 ± 0.17 Å (*r_A_*) and 4.57 ± 0.32 Å (*r_B_*). Conversely, the reference distance H14–H2 exhibited minimal deviation across different conformations, measuring at 4.23 ± 0.01 Å (*r_ref_*) for compound **1** and compound **2**.

The cross-relaxation rates for the experimentally determined and reference distances were previously computed utilizing the IRA. These rates were found to be 1.89 ± 0.08 × 10^−3^ s^−1^ and 7.04 ± 0.25 × 10^−3^ s^−1^ for the H14–H2 and H6–H9 distances in the structure of compound **1**. Similarly, for compound **2**, the corresponding rates were 1.90 ± 0.04 × 10^−3^ s^−1^ and 6.84 ± 0.23 × 10^−3^ s^−1^ for the same H14–H2 and H6–H9 distances (refer to Figure 9).

As the cross-relaxation rates are nearly identical, both for reference and experimentally determined distances in compound **1** and compound **2**, the H6–H9/13 distance values obtained through the ISPA approach (Equation (7)) are expected to exhibit similar behaviors. Experimental measurements yielded H6–H9/13 distances of 3.40 Å and 3.42 Å for compound **1** and compound **2**, respectively (*r_exp_*).

Combining these experimental distances (*r_exp_*) with quantum chemical calculations (*r_A_* and *r_B_*) makes deducing the proportions of conformer groups within the studied compounds feasible. The methodology for determining these proportions was drawn from previous work [66], involving the plotting of the average distance (*r_ij_*) against the proportion of a conformer group (A or B).

In the context of a two-position exchange (A) ↔ (B), this relationship follows a relatively simple form (Figure 10a,c), represented by a nonlinear curve *r_exp_* = f(*P_A_*) (Equation (8)). The experimental distance value (red line) and its associated error range (green lines) are marked along the ordinate axis. The projection of these values onto the curve *r_exp_* = f(*P_A_*) allows for the determination of the proportion of conformer group A, along with its corresponding error range, for both compound **1** (Figure 10a) and compound **2** (Figure 10c).
(8)$$r_{\exp}=\sqrt[{6}]{\frac{r_{A}^{6}r_{B}^{6}}{P_{A}r_{B}^{6}+P_{B}r_{A}^{6}}}\Rightarrow P_{A}=\frac{r_{A}^{6}\left(r_{B}^{6}-r_{\exp}^{6}\right)}{r_{\exp}^{6}\left(r_{B}^{6}-r_{A}^{6}\right)}$$

In the examined groups of conformers (refer to Appendix A), the variables *r_A_* and *r_B_* represent experimental distances. The parameter *r_exp_* signifies the average or effective experimental distance derived from the NOESY experiment. Variable *P_A_* denotes the fraction of conformer groups labeled as A, while *P_B_*, as its complement, designates the proportion of conformer groups identified as B.

Figure 10 illustrates the proportions of conformer groups A and B for compound **1**, indicating that they account for 49% and 51% of the population, respectively. Conversely, compound **2** exhibits a distinct distribution, with conformer groups A and B representing 98% and 2%, respectively. The introduction of a fluorine substituent has the potential to induce alterations in the favored molecular conformation. This phenomenon may be elucidated by considering the dimensions and electronegativity of the fluorine atom, as expounded upon in the work of H-J. Böhm [67]. These findings suggest that the introduction of fluorine induces a shift in the prevailing molecular conformations of the compounds. This alteration may be a critical determinant in the diverse anticancer activities discussed in a previous study [19]. However, more than the presented results may be required to explain the pharmaceutical effects of the considered compounds, since the physiological processes occur in a water-like system. Here, structural analysis is performed based on NMR results in CDCl_3_. Prior research [28,68,69] has emphasized that the conformation of active pharmaceutical ingredient (API) molecules can profoundly influence membrane structure and domain dynamics. Consequently, variations in API conformations may lead to distinct interactions with the cell membrane and result in differing pharmaceutical effects.

## 3. Materials and Methods

The synthesis and characterization of compound **1**, compound **2**, and additional fluorine-substituted derivatives of 1,5-diaryl-3-oxo-1,4-pentadiene were previously conducted and documented in [19]. The chloroform (CDCl_3_) (99.8 atom % D; CAS no. 865-49-6) was acquired from Sigma Aldrich Rus (Moscow, Russia) and utilized without further purification.

The NMR spectra were acquired using a Bruker Avance III 500 spectrometer (Karlsruhe, Germany). The sample temperature was regulated and monitored using BVT-2000 and BCU 05 equipment. The CDCl_3_ samples were made by dissolving 40 mg of material in 1 mL of solvent. The measurements were conducted at 298.15 K (25 °C) and under atmospheric pressure. The NMR spectra were acquired using a collection of pulse programs available in the TopSpin 3.6.1 software.

The ^1^H NMR spectra [70] were captured within the frequency range of 0 to 13 ppm, with eight scans performed. The relaxation period between scans was 3 s, and the number of data points (FID) collected was 16,384. The ^13^C NMR spectra were recorded in the frequency range of 0 to 237 ppm. The experiment had 12,288 scans with a relaxation delay of 3 s and a total of 65,536 data points. Tetramethylsilane was used as an internal standard to calibrate the NMR chemical shift values, with a reference value of δ_TMS_ = 0 ppm [71,72].

The spectra of ^1^H–^13^C HSQC (heteronuclear single quantum coherence spectroscopy) [73,74,75] and ^1^H–^13^C HMBC (heteronuclear multiple bond correlation spectroscopy) [31,76,77] were recorded. The HSQC spectrum is shown in Appendix A [78], while the HMBC spectrum is shown in Appendix A [79]. The spectral window was from 13 ppm to 237 ppm, with 256 data points on the F1 axis and 1024 data points on the F2 axis. The experiment was repeated 32 times. Homonuclear ^1^H–^1^H TOCSY (total correlation spectroscopy) [33,80,81] spectra were obtained at various mixing durations (20 ms and 100 ms) within the spectral range of 13 ppm × 13 ppm. The number of accumulations was 16. The corresponding figures are in the Appendix A (Appendix A for 20 ms and Appendix A for 100 ms) [70]. The ^1^H–^1^H NOESY spectra were obtained in the 13 ppm × 13 ppm. The number of scans performed was 40. These spectra are shown in Figure 3 and may be found in the references [82,83].

The 1D NMR spectra (^1^H, ^13^C) were analyzed using 2D NMR spectroscopy techniques, including ^1^H–^13^C HSQC, ^1^H–^13^C HMBC, and ^1^H–^1^H TOCSY. Therefore, to identify the protons connected to carbon by a single chemical bond, a ^1^H–^13^C HSQC spectroscopy technique was employed. This technique produces two-dimensional spectra that display the chemical shift values of hydrogen on the *x*-axis and carbon on the *y*-axis. The ^1^H–^13^C HMBC data were used to ascertain the connectivity between protons and carbons bonded by more than a single chemical bond. The correlations between hydrogen atoms in a consecutive sequence of interconnected spins, with a distance of 2–5 chemical bonds, were assured using ^1^H–^1^H TOCSY spectra with varying mixing periods. The chemical shifts of the signals in the ^1^H and ^13^C spectra, as well as the identification of the cross-peaks in the ^1^H–^13^C HSQC, ^1^H–^13^C HMBC, and ^1^H–^1^H TOCSY spectra, are provided in Appendix A.

The investigated compounds were subject to quantum chemical calculations to determine their geometric and energy parameters. These calculations were performed using the Gaussian 09 (Version EM64L-G09RevA.01) software program [84]. In this study, the quantum chemical calculations were conducted in a multi-step approach. Initially, the semiempirical Austin Model 1 (AM1) method was employed to assess the geometry of likely conformers and identify low-energy structures. Subsequent optimization was carried out using the Becke, 3-parameter, Lee–Yang–Parr (B3LYP) method in conjunction with the 6-31G and CC-pVTZ basis sets, as suggested by Armstrong et al. [85]. The data analysis enabled us to discern 62 molecular configurations for compound **1** (as shown in Appendix A) and 39 molecular configurations for compound **2** (as shown in Appendix A), distinguished by their distinct structural and energetic properties. The energy disparity among the identified conformers does not exceed 36.35 kJ/mol. The energy of the conformers being considered and the values of the changing angles are provided in Appendix A. The molecular structures of conformers in XYZ format are given in Appendix A.

The quantum chemical computations conducted in this study excluded solvent effects. Prior research [25,86,87] has demonstrated that accounting for solvent effects in quantum chemical calculations can substantially diminish the set of plausible conformers, an undesirable outcome in our investigation, as it can lead to substantial inaccuracies. The primary criterion for establishing the correctness of our obtained conformers is their consistency with experimental data, as demonstrated in our previous work. Incorrect conformer configurations will manifest inconsistencies with the NOESY experiment [87], resulting in physically and chemically unsupported outcomes. Although no conformational inconsistencies with the NOESY experiments were found, we recognize the importance of considering solvent effects in quantum mechanics (QM) calculations.

The approach employed herein does not explicitly consider dispersion correction. While we acknowledge the importance of incorporating dispersion corrections in quantum chemical calculations and structure optimization in some cases, the utilization of techniques advocated by the reviewer, particularly Grimme’s D3 corrections, can introduce significant errors in our case. This is because D2 and D3 corrections rely on neutral atomic parameters for each chemical element [88]. In certain instances involving molecules or halides, such as fluorine, the density-driven errors in uncorrected DFT may surpass the benefits of dispersion corrections. In such situations, alternative approaches like Hartree–Fock density-functional-theory (HF-DFT) may enhance bond energies, while dispersion corrections may prove ineffectual [89].

Furthermore, the BLYP, BPBE, BP86, and PBE functionals, including a D3 dispersion correction, tend to worsen their performance compared to the non-augmented parent functionals [90].

## 4. Conclusions

In conclusion, 1,5-Diaryl-3-Oxo-1,4-Pentadiene derivatives are unique, organic compounds characterized by a pentadiene core with two conjugated double bonds at the first and fifth positions. They feature two aryl groups that enhance stability and reactivity, along with a ketone group at the third carbon atom, influencing their chemical properties. The analysis of these compounds involved the determination of conformer proportions when dissolved in the organic solvent CDCl_3_. This study explored the influence of fluorine on the molecular conformation and observed variations in dihedral angles τ_1_ (C_6_-N_1_-C_7_-C_8_) and τ_2_ (C_2_-N_1_-C_7_-P_1_). The analysis of ^1^H NMR spectra revealed chemical shifts for specific proton signals, which were attributed to the introduction of the fluorine substituent and conformational changes. A fluorine atom in the para position of one of these compounds (compound **2**) leads to significant conformational variations compared to the original compound (compound **1**).

Furthermore, the study utilized NOESY spectra to identify cross-peaks indicating interactions between hydrogen atoms, and these data were used to determine internuclear distances. The reference-free NOE methodology was applied to assess the predominant conformation of these molecules based on cross-relaxation rates. The study found that compound **1** and compound **2** have distinct conformer populations, with compound **2** primarily favoring conformer group A (98%) and compound **1** having a more balanced distribution of conformer groups A (49%) and B (51%). This structural modification could have important implications for their potential pharmaceutical applications and the developing of new dosage forms with improved pharmaceutical characteristics.

## Figures and Tables

**Figure 1 ijms-24-16707-f001:**
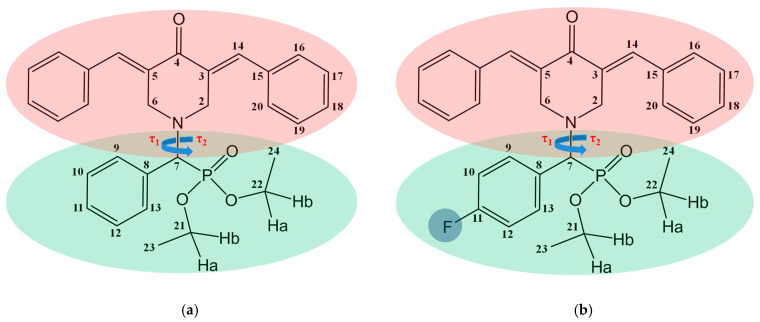
Molecular structure of compound **1** (**a**) and compound **2** (**b**) with designation of dihedral angles τ_1_ (C_6_-N_1_-C_7_-C_8_) and τ_2_ (C_2_-N_1_-C_7_-P_1_) (blue arrows) and corresponding fragments—structures of 3,5-dibenzylidene-4-oxopiperidine (red region) and diethyl(phenyl)methylphosphonate (green region). Atoms numbering is used to label signals and cross-peaks in NMR spectra and designate dihedral angles.

**Figure 2 ijms-24-16707-f002:**
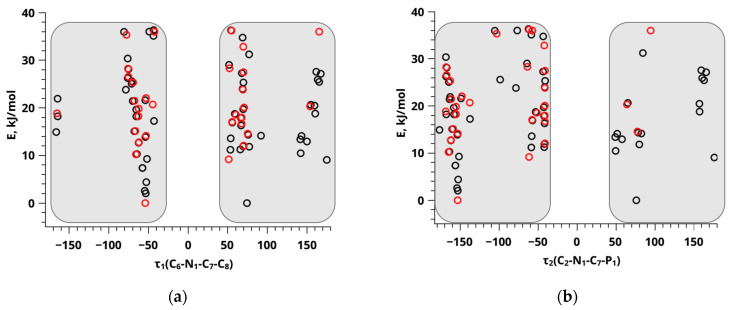
Energy values of different conformers of compound **1** (black dots) and compound **2** (red dots) versus dihedral angles τ_1_ (C_6_-N_1_-C_7_-C_8_) (**a**) and τ_2_ (C_2_-N_1_-C_7_-P_1_) (**b**) obtained using the B3LYP method in conjunction with the 6-31G and CC-pVTZ basis sets. The gray areas indicate the ranges of characteristic values for the respective angles.

**Figure 3 ijms-24-16707-f003:**
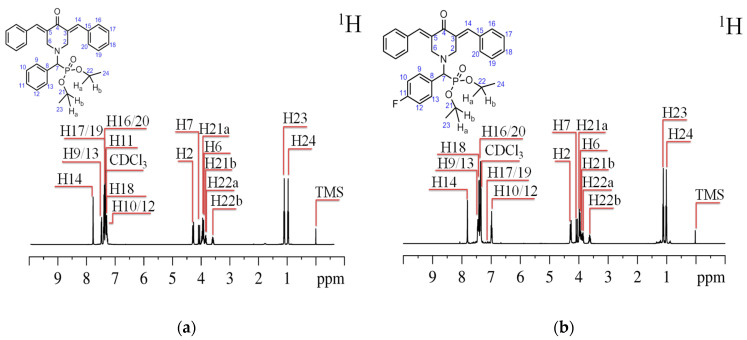
^1^H NMR spectra of compound **1** (**a**) and compound **2** (**b**) in CDCl_3_. The graphic displays the numbering of the atoms, indicated by blue numbers in the upper left portion.

**Figure 4 ijms-24-16707-f004:**
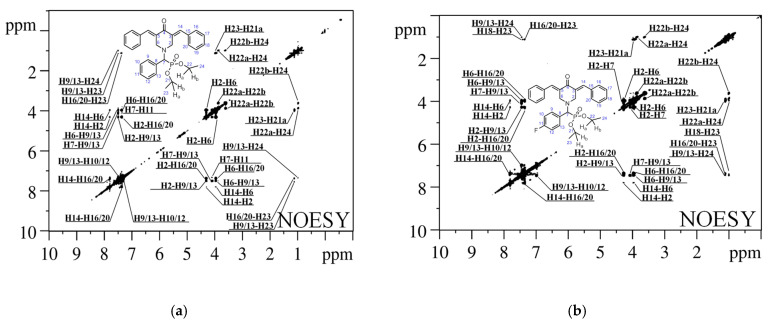
^1^H–^1^H NOESY spectra of compound **1** (**a**) and compound **2** (**b**) in CDCl_3_.

**Figure 5 ijms-24-16707-f005:**
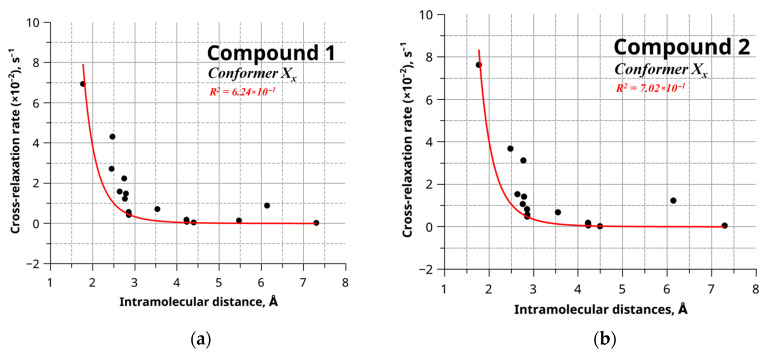
Dependence of the cross-relaxation rate on the internuclear distance for one of the conformers for compound **1** (**a**) and compound **2** (**b**). The red line indicates the approximating curve 1/r^6^.

**Figure 6 ijms-24-16707-f006:**
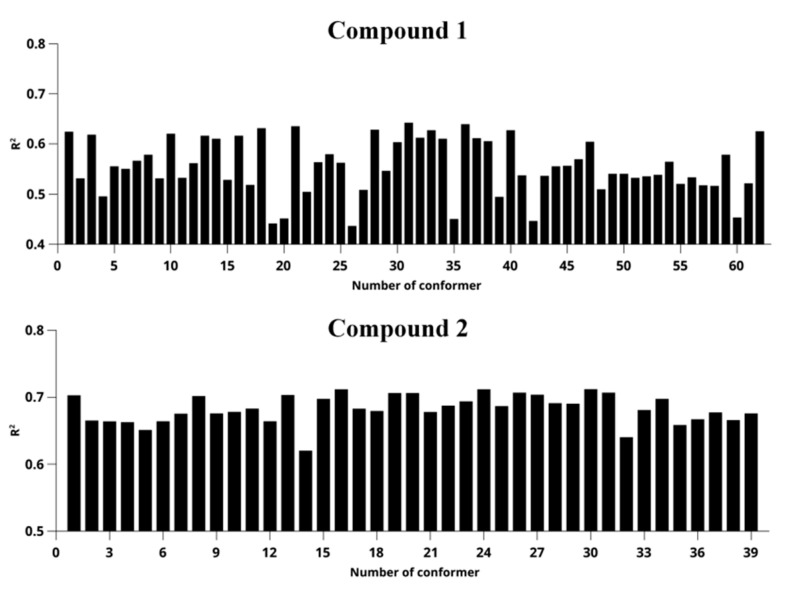
Distribution of coefficients of determination obtained for each conformer of compound **1** (**top**) and compound **2** (**bottom**).

**Figure 7 ijms-24-16707-f007:**
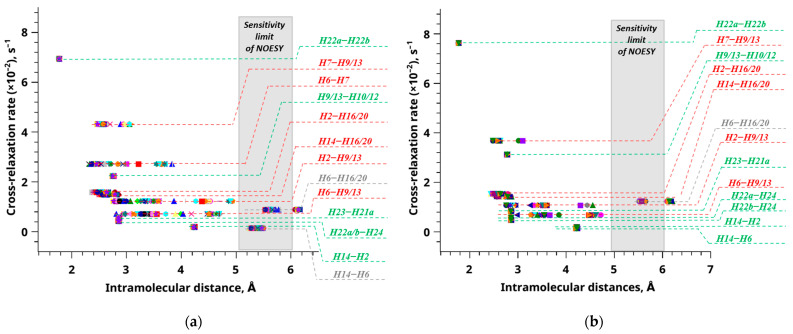
The relationship between the cross-relaxation rate and internuclear distance is depicted for compound **1** (**a**) and compound **2** (**b**) conformers. The shaded region represents the NOESY method’s sensitivity range (5–6 Å), with gray lines denoting distances within this range. Red lines indicate distances influenced by molecular conformation, while green lines represent distances unaffected by molecular conformation. The utilization of various colors in this context serves the purpose of categorizing distinct conformers.

**Figure 8 ijms-24-16707-f008:**
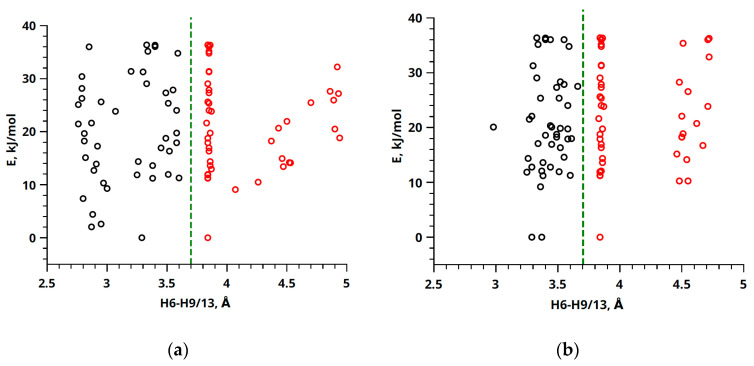
Energy values of different conformers A (black dots) and B (red dots) versus distances H6/H9–H13 for compound **1** (**a**) and compound **2** (**b**) obtained using the B3LYP method in conjunction with the 6-31G and CC-pVTZ basis sets. In this context, utilizing the green line serves the purpose of visually demarcating distinct groups of conformers.

**Figure 9 ijms-24-16707-f009:**
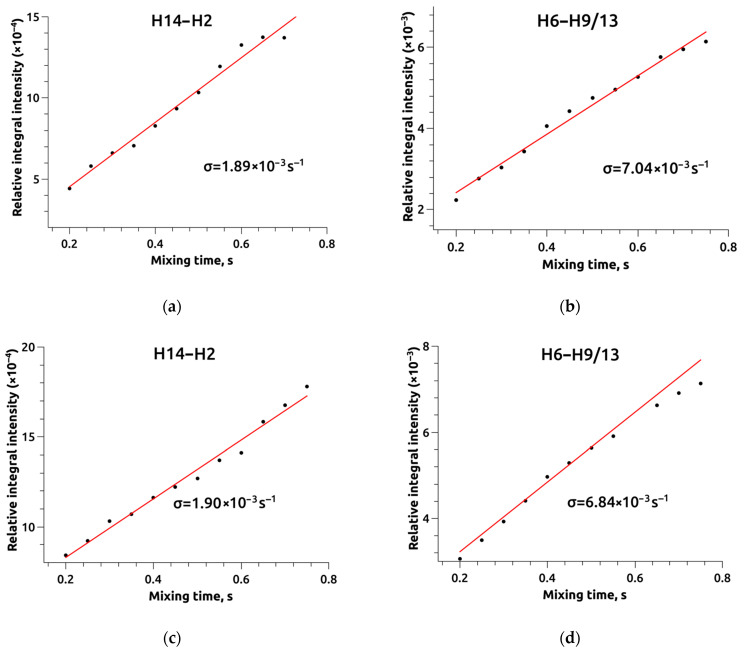
Average integral intensity of H2–H14 distances (**a**,**c**) and H6–H9/13 distances (**b**,**d**) derived from NOESY spectral analysis for compound 1 (**a**,**b**) and compound 2 (**c**,**d**).

**Figure 10 ijms-24-16707-f010:**
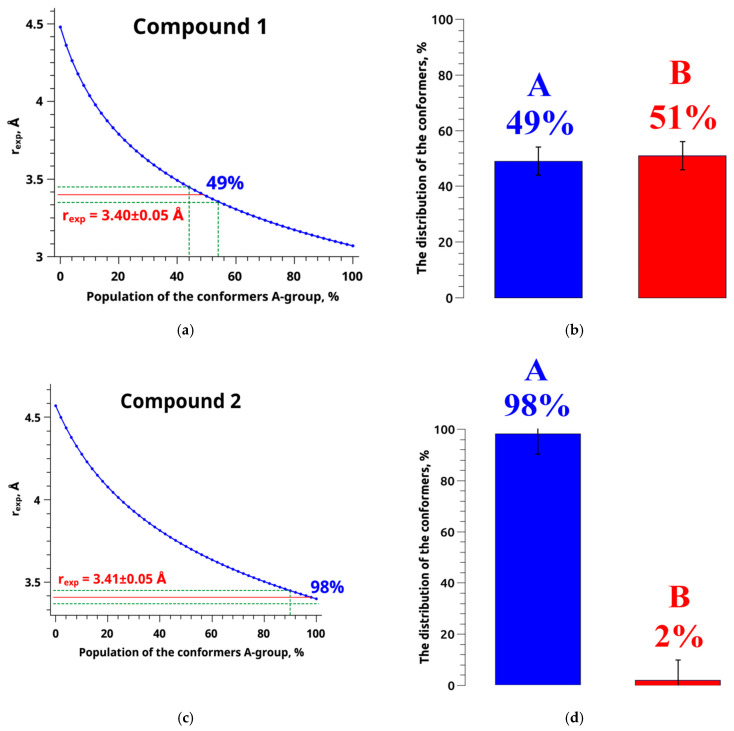
Differential plots depicting the variance between computed and observed distances as functions of conformer populations for compound **1** (**a**) and compound **2** (**c**). In these plots, fractions of conformer A are represented by blue lines, the fractions of conformer B by red lines, and dashed green lines denote distance accuracy thresholds. Experimental data were derived from NOESY spectra recorded in CDCl_3_ for both compound **1** and compound **2**. Tonformer populations of compound **1** (**b**) and compound **2** (**d**) in CDCl_3_ were determined based on NOESY data.

## Data Availability

Data is contained within the article and Appendix A.

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
