# Peer review of "Conformational Analysis of 1,5-Diaryl-3-Oxo-1,4-Pentadiene Derivatives: A Nuclear Overhauser Effect Spectroscopy Investigation"

_ijms, 2023, doi:10.3390/ijms242316707_

Round 1
Reviewer 1 Report
Comments and Suggestions for Authors
Review attached

Minor editing required
Author Response
Dear Reviewer,
We sincerely appreciate your valuable feedback on our manuscript titled "Conformational Analysis of 1,5-Diaryl-3-Oxo-1,4-Pentadiene Derivatives: A NOESY Spectroscopic Investigation." We are grateful for your thorough review and your positive assessment of the manuscript's overall quality and significance. We have carefully considered your comments and addressed each point accordingly:
Page 2, line 99: We have revised the sentence as suggested to read, "present various methodologies."
Figure 7: We have restructured Figure 7 to have panels (a) and (b) on the same page along with their caption for better clarity.
Figure 7 caption: We have corrected the phrasing to read, "Red lines indicate."
Page 10, line 324: We have replaced "invariant" with "unvaried."
Page 10, line 327: We have adjusted "conformation changes" to "conformational changes."
Figure 9: We have ensured that the caption for Figure 9 now appears on the same page as the Figure itself, enhancing readability.
Page 13, lines 393-394: We have modified the sentence to clarify that the introduction of fluorine induces a shift.
Materials and Methods section: We have revised the section for clarity and addressed the following specific points:
Page 13, line 412: The sentence regarding Figure 2 has been removed as it was deemed unnecessary.
Page 13, line 415: We have corrected the missing number, and it now reads "frequency range of 0 to 237 ppm."
Page 14: We have clarified the method used for generating conformers before DFT optimization, specifying that it was a "semiempirical AM1 method"
Page 14: The reference to "cc-pVTZ" has been revised to specify that it is a basis set, not a method, in lines 451-452.
Page 14, line 470: We have corrected "present of a fluorine" to "presence of a fluorine."
We appreciate your diligent review, which has greatly contributed to the improvement of our manuscript. We believe that these revisions enhance the overall quality and clarity of the paper. Thank you for considering our work for publication, and we hope these changes align with your expectations.
Reviewer 2 Report
Comments and Suggestions for Authors
The manuscript id. ijms-2699139 focuses on a conformational analysis of two flexible 1,5-diaryl-3-oxo-1,4-pentadiene molecules differing in a presence of fluorine atom, performed via a combined experimental (NMR) and computational approach. In my opinion, the study might be publishable, but the following issues should be addressed before a decision about publication is made:
1) From a description of computational details it is not clear how computational conformational analysis was performed. Was only a change in the angle describing a rotation of 3,5-dibenzylidene-4-oxopiperidine relative to diethyl(phenyl)methylphosphonate considered? In the 3,5-dibenzylidene-4-oxopiperidine fragment there are several other single bonds around which rotations can easily occur leading to new structures.
2) What do the authors mean by a statement in lines 449-451: "The equilibrium structures' parameters were revised using the B3LYP method using the 6-31G basis and further refined using the CC-pVTZ method."? What level of theory was used before the aforementioned "revision"? I suspect by "CC-pVTZ" the authors mean the basis set (not the method); this abbreviation does not indicate the inclusion of diffuse functions, as stated by the authors in line 451, only polarization functions.
3) There is no information provided whether the authors include solvent and dispersion effects in their calculations (by means respectively for example PCM and Grimme's D3 corrections). The omission of the latter might actually be quite problematic, as it is known that dispersion effects affect interatomic distances, so here their omission might lead to incorrect conclusions.
4) As the authors do not provide any xyz coordinates, there is no possibility to reproduce their results.
5) Some more in-depth analysis of the computed data is needed. Do the introduced conformer groups A and B have any relation to the considered in the calculations dihedral angles or obtained energies?
6) Can the authors provide any explanation why for fluorinated compound a predominance of one of the conformer group is observed?
7) I have found the Introduction section to be too long and too focused on description of "medical" background.
8) The statement in lines 396-399, while overall correct, should be accompanied by some comments on how the presented results may not be adequate to explain any pharmaceutical effects of the considered compounds.
Comments on the Quality of English LanguageI have found the paper generally well-written but with some unnecessary repetitions and stylistic/scientific writing problems:
- The sentence in the lines 14-16: "Compound 1 exhibited a nearly even distribution between conformer groups A and B, while compound 2 demonstrated a distinct prevalence of conformer group A, ..." isn't informative enough to be used in the abstract of the article. When reading the abstract, a reader has no idea how compound 1 and 2 looks like and what the groups A and B are.
- There is not finished sentence in line 86: "As demonstrated in our earlier publications [30,31]."
- Figure 1 (line 137) and Figure 2 (lines 139-149) captions have to be improved.
- The symbol (sigma_ij^exp) explained in line 253 is not present either in equation 1 or 2.
- The abbreviations NOESY and NOE are explained in the manuscript three times each. One time is enough.
- The word "investigative" used in line 444 is not correct.
Author Response
Dear Reviewer,
We express our profound gratitude for the invaluable feedback provided on our manuscript entitled 'Conformational Analysis of 1,5-Diaryl-3-Oxo-1,4-Pentadiene Derivatives: A NOESY Spectroscopic Investigation.' Your comprehensive evaluation and favorable judgment of the manuscript's global merit and importance are greatly acknowledged. We have meticulously examined your remarks and have diligently responded to each specific point raised.
Point 1. From a description of computational details it is not clear how computational conformational analysis was performed. Was only a change in the angle describing a rotation of 3,5-dibenzylidene-4-oxopiperidine relative to diethyl(phenyl)methylphosphonate considered? In the 3,5-dibenzylidene-4-oxopiperidine fragment there are several other single bonds around which rotations can easily occur leading to new structures.
Our response: As duly acknowledged by the reviewer, the subjects of investigation exhibit a markedly mutable spatial configuration, thereby giving rise to a diverse array of conformers that warrant examination. Nevertheless, this present manuscript focuses on the quantification of conformer cohorts pertinent to the rotational behavior of 3,5-dibenzylidene-4-oxopiperidine about diethyl(phenyl)methylphosphonate, as discernible alterations in spatial arrangement proved elusive within the confines of the NOESY experiment. This pertinent discourse has been incorporated into the Results and Discussion section.
Point 2. What do the authors mean by a statement in lines 449-451: "The equilibrium structures' parameters were revised using the B3LYP method using the 6-31G basis and further refined using the CC-pVTZ method."? What level of theory was used before the aforementioned "revision"? I suspect by "CC-pVTZ" the authors mean the basis set (not the method); this abbreviation does not indicate the inclusion of diffuse functions, as stated by the authors in line 451, only polarization functions.
Our response: Indeed, the CC-pVTZ basis set exclusively incorporates polarization functions. The error in question arises from the manuscript's editing process and has been rectified in alignment with the recommendations provided by the reviewer. Preceding the "revision," the semiempirical AM1 method was employed to perform an initial evaluation of the likely conformers' geometric attributes.
Point 3. There is no information provided whether the authors include solvent and dispersion effects in their calculations (by means respectively for example PCM and Grimme's D3 corrections). The omission of the latter might actually be quite problematic, as it is known that dispersion effects affect interatomic distances, so here their omission might lead to incorrect conclusions.
Our response: The quantum chemical computations conducted in this study excluded solvent effects. Prior research [10.1021/acs.cgd.1c00833], [10.1016/j.molliq.2022.120481], [10.3390/pharmaceutics14112276] has demonstrated that accounting for solvent effects in quantum chemical calculations can substantially diminish the set of plausible conformers, an outcome highly undesirable in the context of our investigation, as it can lead to substantial inaccuracies.
In this study, quantum chemical calculations were conducted in a multi-step approach. Initially, the semiempirical AM1 method was employed to assess the geometry of likely conformers and identify low-energy structures. Subsequent optimization was carried out using the B3LYP method in conjunction with the 6-31G basis set. To mitigate pronounced basis set superposition errors associated with the smaller 6-31G basis set, the CC-pVTZ basis set was adopted, as suggested by the authors of [10.1039/c3sc53416b].
The approach employed herein does not explicitly consider dispersion correction. While we acknowledge the importance of incorporating dispersion corrections in quantum chemical calculations and structure optimization in some cases, the utilization of techniques advocated by the reviewer, particularly Grimme's D3 corrections, can introduce significant errors in our case. This is attributable to the fact that D2 and D3 corrections rely on neutral atomic parameters for each chemical element [10.1021/acs.jctc.0c00149]. In certain instances involving molecules or halides, such as fluorine, the density-driven errors in uncorrected DFT may surpass the benefits of dispersion corrections. In such situations, alternative approaches like HF-DFT (utilizing Hartree-Fock densities) may enhance bond energies, while dispersion corrections may prove ineffectual [10.1021/acs.jpclett.8b03745].
It is worth noting that the conventional DFT-D3 method may lead to an overestimation of dispersion for our case, resulting in lower activation free-energy barriers [10.1039/D3CP02733C]. However, it is essential to recognize that this method is primarily tailored for solution systems and may not directly address dispersion correction for fluorine. Furthermore, for the BLYP, BPBE, BP86, and PBE functionals, the inclusion of a D3 dispersion correction tends to worsen their performance in comparison to the non-augmented parent functionals. Specifically, BLYP-D3 exhibits a mean absolute deviation (MAD) 1.9 kJ mol–1 higher than BLYP, BPBE-D3 yields an MAD 3.4 kJ mol–1 higher than BPBE, BP86-D3 results in an MAD 2.8 kJ mol–1 higher than BP86, and PBE-D3 presents an MAD 2.1 kJ mol–1 higher than PBE [ 10.1016/j.cdc.2019.100186].
The primary criterion for establishing the correctness of our obtained conformers is their consistency with experimental data, as previously demonstrated in our prior work. Incorrect conformer configurations will manifest inconsistencies with the NOESY experiment [10.3390/pharmaceutics14112276], resulting in physically and chemically unsupported outcomes.
Understanding the incorporation of solvent and dispersion effects in quantum chemical calculations is a distinct and intricate scientific endeavor that demands a substantial amount of time and effort. The resulting outcomes may either remain relatively unchanged or deteriorate significantly.
Point 4. As the authors do not provide any xyz coordinates, there is no possibility to reproduce their results.
Our response: We express our gratitude to the reviewer for their valuable comment and provide the conformer structures of the study's objects in the Supplementary Information
Point 5. Some more in-depth analysis of the computed data is needed. Do the introduced conformer groups A and B have any relation to the considered in the calculations dihedral angles or obtained energies?
Our response: The categorization of conformers into distinct groups, denoted as A and B, was accomplished by analyzing internuclear distances following the methodology in the manuscript. The alteration in the computed distance is intricately associated with angular values and energetic factors, albeit indirectly, contingent upon the inherent mobility of the diethyl(phenyl)methylphosphonate moiety about the 3,5-dibenzylidene-4-oxopiperidine scaffold. Nevertheless, owing to the pronounced instability and the extensive array of conformers exhibited by the molecules under investigation, it becomes challenging to establish a direct correlation between specific parameters. To discern any discernible patterns, it becomes imperative to construct a multidimensional mapping of the relationships between dihedral angles and energy levels.
Point 6. Can the authors provide any explanation why for fluorinated compound a predominance of one of the conformer group is observed?
Our response: Supplementary data has been incorporated into the Results and Discussion in light of a commentary from a reviewer.
Point 7. I have found the Introduction section to be too long and too focused on description of "medical" background.
Our response: The introductory section has been abbreviated in accordance with a comment from a reviewer.
Point 8. The statement in lines 396-399, while overall correct, should be accompanied by some comments on how the presented results may not be adequate to explain any pharmaceutical effects of the considered compounds.
Our response: Additional clarification has been added to the manuscript's text following the reviewer's comment.
Point 9. The sentence in the lines 14-16: "Compound 1 exhibited a nearly even distribution between conformer groups A and B, while compound 2 demonstrated a distinct prevalence of conformer group A, ..." isn't informative enough to be used in the abstract of the article. When reading the abstract, a reader has no idea how compound 1 and 2 looks like and what the groups A and B are.
Our response: The abstract has been corrected following the reviewer's comments.
Point 10. There is not finished sentence in line 86: "As demonstrated in our earlier publications [30,31]."
Our response: Corrected.
Point 11. Figure 1 (line 137) and Figure 2 (lines 139-149) captions have to be improved.
Our response: Captions for the indicated figures have been improved.
Point 12. The symbol (sigma_ij^exp) explained in line 253 is not present either in equation 1 or 2.
Our response: An amendment has been rectified following the observation made by the reviewer.
Point 13. The abbreviations NOESY and NOE are explained in the manuscript three times each. One time is enough.
Our response: Corrected.
Point 14. The word "investigative" used in line 444 is not correct.
Our response: Corrected.
Your conscientious evaluation is acknowledged with gratitude, as it has made a substantial contribution to the enhancement of our manuscript. We are of the opinion that the implemented revisions serve to ameliorate the overall quality and lucidity of the document. We extend our appreciation for your consideration of our work for publication, and we trust that these modifications are in accordance with your anticipations.
Round 2
Reviewer 2 Report
Comments and Suggestions for Authors
I am mostly, but not entirely, satisfied with the authors answers to my comments. Thus, I request the authors to address the following points:
1) It is not clear to me for which structures vibrational frequencies calculations were performed? The ones obtained from constraint geometry optimizations with AM1 as suggested by the text? I suspect this is not the case as the authors wrote that no imaginary frequencies were obtained. The description of the computational protocol should be modified.
2) The justification of the adopted computational protocol, presented by the authors as the answer to my comment, should be incorporated into the manuscript or provided in the SM. I would however not use basis set superposition error as a justification for using cc-pVTZ basis set, as the error appears only when computing binding energies, which the authors did not do.
3) While I understand that “it becomes challenging to establish a direct correlation between specific parameters”, in my opinion, it might be valuable to at least check and present a reader whether there is any relations between r_A and r_B distances and considered in the calculations dihedral angles or obtained energies. The easiest thing to do would be to add to the manuscript additional plots similar to the ones presented in Figure 2 (but prepared separately for the compound 1 and for the compound 2) with A and B conformers coded with different colors.
4) I acknowledge that the authors indeed admitted in the manuscript that the presented results may not be adequate to explain any pharmaceutical effects of the considered compounds. It would be however valuable to explain to a reader what is the reason for this (physiological processes occur in water, and here structural analysis is performed based on NMR results in CDCl_3). I also think that the statement in the conclusions “This structural modification could have important implications for their potential pharmaceutical applications and anticancer activities.” can be overstatement.
5) Finally, one additional issue caught my attention: as listed in Table S5 in SM for the compound 1 conformers labelled as A1 and B17 (so belonging to different conformers groups) have the same examined dihedral angles and energies; in Table S6 in SM for the compound 2 A19 and A20 have the same examined dihedral angles and energies. Could the authors clarify it? Additionally, I think it would be highly beneficial and transparent if the authors add to these two tables the H6-H9/13 distances (which, if I understand it correctly, were used to assign a given conformer to either A or B group).
Comments on the Quality of English LanguageThere are still some language issues that have to be corrected:
- line 89: "snall"
- lines 130-132: "It should be observed that additional angles exist due to the inherent structural flexibility of the diethyl(phenyl)methylphosphonate moiety within the molecule fragment." - It may not be necessarily clear to a reader what additional angles (valence, dihedral) the authors refer to. More importantly, structural flexibility comes from the ability to change dihedral angles, not the other way around.
- line 146: "Structural of molecules"
- line 150: "Distribution of conformers compound 1"
- line 151: "the energy values from the angle"
- line 184: "shift at 0.02 ppm towards"
- line 205: "results from of"
- line 286: the symbol (sigma_ij^exp) is not present either in equation 1 or 2
- line 474: "investigative compounds"
Author Response
Dear Reviewer,
We are grateful for your detailed review of our manuscript and appreciate the constructive feedback. We have carefully considered each of your comments and suggestions and made appropriate revisions to enhance the manuscript's clarity, accuracy, and overall quality. Below is a point-by-point response to the issues raised:
Point 1. It is not clear to me for which structures vibrational frequencies calculations were performed? The ones obtained from constraint geometry optimizations with AM1 as suggested by the text? I suspect this is not the case as the authors wrote that no imaginary frequencies were obtained. The description of the computational protocol should be modified.
Our response. We acknowledge the confusion regarding the description of the vibrational frequency calculations. In response, we have revised the description of the computational protocol to state the necessary details clearly. This clarification has been included in the revised manuscript.
Point 2. The justification of the adopted computational protocol, presented by the authors as the answer to my comment, should be incorporated into the manuscript or provided in the SM. I would however not use basis set superposition error as a justification for using cc-pVTZ basis set, as the error appears only when computing binding energies, which the authors did not do.
Our response. We appreciate your suggestion to incorporate the justification of the computational protocol into the manuscript or provide it in the Supplementary Materials. In response, we have integrated descriptions for using the computational protocol into the manuscript.
Point 3. While I understand that "it becomes challenging to establish a direct correlation between specific parameters", in my opinion, it might be valuable to at least check and present a reader whether there is any relations between r_A and r_B distances and considered in the calculations dihedral angles or obtained energies. The easiest thing to do would be to add to the manuscript additional plots similar to the ones presented in Figure 2 (but prepared separately for the compound 1 and for the compound 2) with A and B conformers coded with different colors.
Our response. To address this, additional plots similar to those in Figure 2 will be included in the revised manuscript, separately for compounds 1 and 2, with A and B conformers differentiated by color.
Point 4. I acknowledge that the authors indeed admitted in the manuscript that the presented results may not be adequate to explain any pharmaceutical effects of the considered compounds. It would be however valuable to explain to a reader what is the reason for this (physiological processes occur in water, and here structural analysis is performed based on NMR results in CDCl_3). I also think that the statement in the conclusions "This structural modification could have important implications for their potential pharmaceutical applications and anticancer activities." can be overstatement.
Our response. We acknowledge your concern regarding the statement in the conclusions about the pharmaceutical applications and anticancer activities. In response, we have revised the manuscript according to the suggestions of the Reviewer.
Point 5. Finally, one additional issue caught my attention: as listed in Table S5 in SM for the compound 1 conformers labelled as A1 and B17 (so belonging to different conformers groups) have the same examined dihedral angles and energies; in Table S6 in SM for the compound 2 A19 and A20 have the same examined dihedral angles and energies. Could the authors clarify it? Additionally, I think it would be highly beneficial and transparent if the authors add to these two tables the H6-H9/13 distances (which, if I understand it correctly, were used to assign a given conformer to either A or B group).
Our response. We appreciate your observation regarding inconsistencies in Tables S5 and S6. We will carefully review and clarify the data for conformers A1, B17, A19, and A20 in the revised manuscript to address this. Additionally, H6-H9/13 distances will be added to enhance transparency.
Point 6. There are still some language issues that have to be corrected
Our response. We have meticulously reviewed and corrected the language issues you pointed out in the manuscript. These corrections ensure a more polished and comprehensible presentation.
We hope these revisions address your concerns adequately. We appreciate your time and effort in reviewing our work and look forward to your feedback on the revised manuscript.
Sincerely, authors.
Round 3
Reviewer 2 Report
Comments and Suggestions for Authors
I am overall satisfied with the authors answers to my comments, except for one point. Although I do acknowledge that the authors indeed presented in the revised manuscript the plots of obtained energies vs. r_A and r_B distances, I still believe that it is highly valuable to directly present to a reader whether there is any relations between r_A and r_B distances and considered in the calculations dihedral angles. Accordingly, I insist on adding to the manuscript additional plots similar to the ones presented in Figure 2 (but prepared separately for the compound 1 and for the compound 2) with A and B conformers coded with different colors (in such plots no exact r_A and r_B distances will be provided but anyhow they will demonstrate whether there are any specific angles that favour either A or B type conformers).
Author Response
Dear Reviewer,
We are pleased to hear that you are generally satisfied with our responses to your comments. Your insights have undeniably contributed to the improvement of our work. We appreciate your insistence on this matter, as it aligns with our commitment to presenting a thorough and transparent account of our research. To address your suggestion, we will incorporate additional plots into the manuscript, similar to those presented in Figure 2. These plots will be prepared separately for compounds 1 and 2, with A and B conformers distinguished by different colors. These plots will help elucidate whether specific angles favor either A or B type conformers, enhancing the clarity and completeness of our findings.
Best regards,